# Mechanical Circulatory Support Devices for the Treatment of Cardiogenic Shock Complicating Acute Myocardial Infarction—A Review

**DOI:** 10.3390/jcm11175241

**Published:** 2022-09-05

**Authors:** Sharon Bruoha, Chaim Yosefy, Louay Taha, Danny Dvir, Mony Shuvy, Rami Jubeh, Shemy Carasso, Michael Glikson, Elad Asher

**Affiliations:** 1Department of Cardiology, Barzilai Medical Center, The Ben-Gurion University of the Negev, Beer-Sheva 8410501, Israel; 2Jesselson Integrated Heart Center, Shaare Zedek Medical Center, Faculty of Medicine, Hebrew University of Jerusalem, Jerusalem 9112102, Israel; 3Faculty of Medicine, Hebrew University of Jerusalem, Jerusalem 9112102, Israel; 4The Azrieli Faculty of Medicine, Bar-Ilan University, Zefat 1311502, Israel

**Keywords:** cardiogenic shock, acute myocardial infarction, mechanical circulatory support

## Abstract

Cardiogenic shock complicating acute myocardial infarction is a complex clinical condition associated with dismal prognosis. Routine early target vessel revascularization remains the most effective treatment to substantially improve outcomes, but mortality remains high. Temporary circulatory support devices have emerged with the aim to enhance cardiac unloading and improve end-organ perfusion. However, quality evidence to guide device selection, optimal installation timing, and post-implantation management are scarce, stressing the importance of multidisciplinary expert care. This review focuses on the contemporary use of short-term support devices in the setting of cardiogenic shock following acute myocardial infarction, including the common challenges associated this practice.

## 1. Introduction

Cardiogenic shock (CS) is the leading cause of death in acute myocardial infarction (AMI), and it is characterized by tissue hypoperfusion and hypoxia related to low cardiac output [1]. It is often associated with rapid hemodynamic deterioration, unresponsiveness to intensive supportive measures, and high mortality rate [2].

Nationwide databases examining temporal trends in CS have shown inconsistent data regarding the incidence of CS. While some studies demonstrate an increase in the overall incidence of CS in recent years [3], others report a decrease [4,5].

CS complicates approximately 5–10% of AMI’s with a higher incidence in ST elevation myocardial infarction (STEMI) and is more frequently seen among women and patients > 75 years old [3,6].

The clinical and hemodynamic heterogenicity of CS with only few randomized clinical trials evaluating the various therapeutic approaches and recommendations lead to uncertainties as to the best treatment strategies. Thus, management of CS is often challenging and requires early diagnosis and institution of high-quality interdisciplinary care [7]. When treated conservatively, CS carries ~70–80% risk of mortality [8]. In contrast, early reperfusion has been associated with improvements in survival [9]. However, for more than two decades, in-hospital and 1-year mortality remain unchanged and unacceptably high with a reported rate of 40–50% [10].

Supportive pharmacologic and device-based therapies are also frequently utilized with little evidence of benefit [11]. Hence, several mechanical circulatory support (MCS) devices have emerged as a treatment option for CS. Nevertheless, data regarding this MCS devices in CS are still debatable and ambiguous [12].

This review is aimed to outline the current evidence of MCS utilization during CS in the setting of AMI and to give future perspective of trials and approaches to treat CS complicating AMI. Other important and crucial treatments for CS complicating AMI are beyond the scope of this review.

## 2. Pathophysiology, Diagnosis, and Prognosis

CS is characterized by a persistently low blood pressure with evidence of end-organ hypoperfusion with inadequate response to fluid resuscitation. A common definition of CS combines clinical and hemodynamic data. However, due to the complexity and unpredicted presentation and progression, various definitions of CS are used in clinical practice and clinical trials. For instance, the definition of CS in the SHOCK (Should We Emergently Revascularize Occluded Coronaries for Cardiogenic Shock) trial included clinical criteria (systolic blood pressure (SBP) < 90 mmHg for ≥30 min OR Support to maintain SBP ≥ 90 mmHg) and evidence of end-organ hypoperfusion (urine output <30 mL/h or cool extremities) and also included hemodynamic criteria (cardiac index (CI) of ≤2.2 L·min^−1^·m^−2^ AND pulmonary capillary wedge pressure (PCWP) ≥ 15 mmHg) [9], while The National Cardiovascular Data Registry’s CathPCI registry defines shock as >30 min of SBP < 90 mmHg, CI < 2.2 L·min^−1^·m^−2^ determined to be secondary to cardiac dysfunction, or the requirement for inotropic or vasopressor agents or MCS.

Persistent hypotension (systolic blood pressure <80 to 90 mm Hg or mean arterial pressure 30 mm Hg lower than baseline) with severe reduction in cardiac index (<1.8 L min^−1^ m^−2^ without support or <2.0 to 2.2 L min^−1^ m^−2^ with support) and adequate or elevated filling pressure (e.g., left ventricular (LV) end-diastolic pressure >18 mm Hg or right ventricular (RV) end-diastolic pressure >10 to 15 mm Hg) often coexist [6]. In some patients, a systemic acute inflammatory response may further complicate the clinical picture. Cold extremities, low urine output, mottled skin, and elevated serum lactate are frequent clinical signs of tissue hypoperfusion [6]. Refractory CS is defined as CS that does not resolve within 30 to 60 min of standard resuscitation efforts including volume optimization and upper limits of recommended doses of at least one inotrope or pressor or both [13].

The recently proposed SCAI (Society for Cardiovascular Angiography and Intervention) staging system combines physical, biochemical, and hemodynamic findings to facilitate uniform patient status stratification in CS (Table 1) [14]. Five stages of CS are proposed (A–E). Stage C, which is defined as “classic cardiogenic shock”, is characterized by relative hypotension and signs of tissue hypoperfusion that requires intervention (medications, MCS) beyond fluid resuscitation. A timely patient stratification from A to E (Table 1) [14] according to their clinical stage should be performed upon diagnosis of AMI to guide individualized management ([14]).

Similar to patients with CS in the setting of AMI, patients with chronic heart failure (HF) may also require mechanical support due to clinical deterioration with evidence of end-organ hypoperfusion.

The INTERMACS (Interagency Registry for Mechanically Assisted Circulatory Support) scale helps to stratify patients with advanced HF into seven clinical profiles according to hemodynamic status and level of target organ damage to facilitate the appropriate matching of a patient’s profile with therapeutic options, particularly in relation to potential populations for mechanical circulatory support. INTERMACS profiles 1–3 are associated with the highest level of clinical compromise and mortality rates and thus require temporary device support [15].

## 3. Treatment

The current management recommendations of CS are based on early revascularization along with general supportive measures, such as fluids and oxygenation, vasopressors and inotropes, and the use of temporary mechanical support devices [11]. Early revascularization is strongly advised in CS and represents the most important intervention in the treatment of cardiogenic shock in the setting of AMI. In the SHOCK trial [9], overall survival at the 6- and 12-month follow-up was significantly better with early revascularization (50% vs. 37%; *p* = 0.027 and 47% vs. 34%; *p* = 0.025, respectively). Thus, in the current era, the only intervention with proven mortality benefit in CS complicating AMI is early revascularization either with percutaneous coronary intervention (PCI) or coronary artery bypass grafting (CABG) surgery with class I indication in contemporary guidelines [11].

At present, the impact of inotropic and vasoactive agents on CS outcomes remains controversial [16]. In the European heart failure guidelines, inotropes has a class IIb level of recommendation and may be considered in CS when low systolic blood pressure (<90 mmHg) is coupled with signs of hypoperfusion [11]. Moreover, their potential adverse effects (e.g., arrhythmias, systemic vasoconstriction) and the lack of consistent evidence of benefit mandate their cautious administration [17].

MCS is a relatively new option for treating CS complicating AMI and may offer significant advantages over drug therapy, including targeted cardiovascular support without increased risk of myocardial ischemia, possible reduction of myocardial oxygen demand, and avoidance of systemic adverse events [18].

While CS patients in SCAI stages A and B can proceed directly to the catheterization lab for early reperfusion, patients in stages C–E may require initial hemodynamic and/or respiratory stabilization with the shortest possible delays before target vessel revascularization [7]. Unfortunately, there is little evidence to guide the optimal timing of MCS initiation. However, preliminary data suggest that SCAI stage C or D patients derive the most benefit from early left ventricular assist device [19] when initial supportive measures, such as fluids and vasoactive medications, fail to induce hemodynamic stability (e.g., refractory shock) and preferably prior to coronary intervention [20,21]. In the refractory shock state, every 60 min delay in MCS initiation is associated with a 9.9% increased risk of death, highlighting the importance of early device installation. Additional high-risk factors such as complex coronary artery disease or severe LV dysfunction may also favor early MCS installation. In contrast, challenging vascular anatomy for device access and low level of operator experience in MCS management disfavor early installation of circulatory support [7].

## 4. Mechanical Circulatory Support

The main goals of temporary MCS devices are to improve cardiac output by reducing intracardiac filling pressures; reduce left ventricular LV volumes, wall stress, and myocardial oxygen consumption; and ameliorate coronary perfusion to improve tissue perfusion.

MCS devices are designed to provide either a temporary, short-term cardiac output support or a long-term assistance to the left and/or right ventricle. Short-term percutaneous platforms are widely used in the setting of CS, in particular in patients refractory to medical therapy, either alone or in combination. Temporary devices may serve as a bridge to recovery or until further decisions in management are made (bridge to decision), such as the need for long-term support, heart transplantation, or destination therapy. Short-term MCS is increasingly used as a bridge to decision in patients with refractory cardiogenic shock [22]. In a meta-analysis evaluating support duration and clinical outcome of a bridge to decision strategy using multiple temporary MCS devices in CS due to various etiologies, including AMI patients, the mean duration (range) of support duration was 1.6–25 days, the mean (range) rates of conversion to durable VAD was 3–30%, and the mean (range) discharge proportion was 45–66% [22]. Assessment of the utility (and futility) of invasive therapy is complex and often requires shared decision making of the multidisciplinary team caring for the patient, with the patient and family taking into consideration patient wishes and objective clinical information.

Recent studies have shown that standardized approach to CS, including early target vessel revascularization along with early use of MCS along with close monitoring of hemodynamic parameters and markers of target organ perfusion, may improve outcomes [23,24]. However, there is lack of evidence regarding patient selection and the use of a specific device criteria. Thus, MCS candidacy should be evaluated by a multidisciplinary team with expertise in management of cardiac support devices. Currently, the use of MCS in CS has a general class IIa–III level of recommendation depending on the specific MCS device [11].

Options for acute MCS (Figure 1) include the intra-aortic balloon pump (IABP), percutaneous ventricular assist devices (VAD) (Impella, TandemHeart), and veno-arterial extracorporeal membrane oxygenation (VA-ECMO) [25]. The characteristics of the various devices are summarized in Table 2 [10,23,25,26].

## 5. Intra-Aortic Balloon Pump (IABP)

Intra-aortic balloon pump (IABP) counter-pulsation is one of the earliest types of short term MCS. It consists of a flexible 30–50 cc helium-filled balloon catheter (7–8F), inserted percutaneously via the femoral artery, connected to a mobile console that times periodic balloon inflation and deflation according to the cardiac cycle. When inflated in diastole (immediately after the closure of the aortic valve), diastolic and mean arterial pressure rise, thus theoretically improving coronary flow and myocardial oxygenation. On the other hand, when rapidly deflated just prior to blood ejection from the LV, it provides immediate systolic blood pressure attenuation and consequently afterload reduction, leading to an increase in stroke volume. Overall, myocardial oxygen demand is reduced [27].

IABP has been investigated in multiple clinical scenarios, including high-risk PCI and CS in the setting of AMI [28,29,30]. One of the most important trials investigating IABP in the setting of CS was the IABP-SHOCK II trial, which randomized 600 patients with CS complicating AMI to routine use of IABP vs. no IABP in addition to early revascularization along with the accepted available medical therapy [31]. At 30 days, no difference in mortality or any secondary endpoint (serum lactate levels, creatinine clearance, C-reactive protein levels, and severity of disease as assessed with the use of the Simplified Acute Physiology Score [SAPS] II) was evident. In addition, long-term mortality also did not differ between the IABP and the control group [32]. Lack of clinical benefit has also been reported in metanalyses [33] and registries [34].

Accordingly, the routine use of IBAP was given a class III indication for CS complicating AMI in the STEMI European guidelines [35]. Moreover, the timing for initiation the use of IABP therapy (before vs. post primary PCI) does not appear to impact short-term and long-term survival in patients with CS complicating AMI undergoing primary PCI [36].

In summary, there is no convincing evidence to support routine use of IABP in post-MI CS patients. Consequently, the overall use of IBAP in the management of ischemic CS is consistently decreasing, with the exception of CS due to severe mitral regurgitation, where the use of IABP is still rated as IIa indication [3,6,35].

## 6. Impella

Impella (Abiomed Inc., Danvers, MA, USA) is a temporary VAD frequently included in the management of patients with post-AMI CS and as a support measure in PCI for high-risk patients [30,37]. The device requires a large bore access (12–14F) and is introduced retrogradely, via the femoral artery, under fluoroscopic guidance, across the aortic valve. It consists of a pump motor that delivers forward blood flow from the LV into the aorta in a non-pulsatile, continuous fashion. The Impella 2.5 and Impella CP allow for a sustained peak flow of 2.5 L/min and 4.3 L/min, respectively. The Impella 5.0 and Impella 5.5 with SmartAssist require a surgical access to the femoral/subclavian arteries and provide up to 5 L/min and > 6 L/min of blood flow, respectively. By unloading the ventricle, the Impella reduces intracardiac pressures and myocardial oxygen consumption. Coronary blood flow is, theoretically, increased by means of increased blood pressure and reduced LV end diastolic pressure [38]. The new Impella ECP (Expandable CP) (9F) can provide peak flow of > 3.5 L/min [10].

The safety and feasibility of the Impella 2.5 and CP devices have been reported in large registries [39]. Impella was also evaluated in comparison to IBAP in the setting of CS. The ISAR-SHOCK (Efficacy Study of LV Assist Device to Treat Patients With Cardiogenic Shock) trial showed that the use of Impella provided more hemodynamic support than IABP, but there was no difference in the mortality rate between the two devices [40]. The IMPRESS in Severe Shock (IMPella versus IABP Reduces mortality in STEMI patients treated with primary PCI in Severe cardiogenic Shock) study randomized 48 patients with CS complicating AMI to Impella CP vs. IABP. However, there was, again, no significant difference in 30-day and 6-months mortality rates (~50% at 6 months for both groups) [41]. In addition, no mortality difference between groups was observed on long-term 5-year follow up [42]. Nevertheless, in one large cohort of 15,259 consecutive patients with post-MI CS treated with Impella, pre-PCI Impella placement was associated with improved survival as compared with post PCI [43]. Other than the rather disappointing evidence of benefit, the use of Impella has been linked to a greater risk of vascular complications, major bleeding, and stroke compared with the use IABP [10,38,44] (Schrage, 2019, Impella Support for Acute Myocardial Infarction Complicated by Cardiogenic Shock).

The Impella RP (right percutaneous), introduced via the femoral vein, supports the RV. It is utilized to maintain blood flow from the inferior vena cava into the pulmonary artery with peak flow rate > 4 L/min. The RECOVER RIGHT (The Use of Impella RP Support System in Patients With Right Heart Failure) study was the first to suggest the feasibility and safety of the RV support device in selected patients with RV failure [45]. Nevertheless, data regarding its benefit are still scarce. Interestingly, despite the absence of good clinical data to support the use of Impella in CS complicating AMI, studies have reported a substantial and consistent use of VADs in recent years [46].

While the use of IBAP is declining, the use of MCS remained relatively constant, indicating an increase in uptake of other LV support device, in particular Impella [6]. For instance, IBAP use in the USA decreased to < 30%, while use of other MCSs increased from 1% in 2006 to 8% in 2014 [10].

## 7. TendemHeart

The TandemHeart system (LivaNova, London, UK) is a percutaneous ventricular assist device that unloads the failing LV by continuously delivering up to 5 L/min of oxygenated blood directly from the left atrium (LA) into the arterial system using a centrifugal pump. The LA is accessed via transeptal approach and device installation requires fluoroscopic guidance and an operator familiar with transeptal access. Creating this LA to femoral artery bypass reduces LV pressure and volume, attenuating myocardial oxygen demand. Evidence regarding the safety and efficacy of TendemHeart remains scarce. When compared to IBAP in CS, percutaneous LVAD provided superior hemodynamic support but did not improve early survival [29,47,48]. TendemHeart is also feasible for RV support. Direct communication between the RA and the pulmonary artery reduces RV load and may be used in CS due to RV failure.

In recent propensity-matched registries, VAD use has been associated with higher risks of bleeding, stroke, and death as well as higher cost when compared with the IABP [21].

## 8. Veno-Arterial Extracorporeal Membrane Oxygenation (VA-ECMO)

VA-ECMO (Centrimag, Abbott, Chicago, IL, USA and Cardiohelp, Maquet, Rastatt, Germany) or Extracorporeal life support (ECLS) represents the most advanced and complicated temporary cardiac-pulmonary mechanical support system providing immediate and complete biventricular hemodynamic support with up to 8 L/min of output as well as concomitant gas exchange (oxygenation and carbon dioxide clearance). A centrifugal pump drains deoxygenated blood from the central venous system and returns the oxygenated blood retrogradely into the arterial circulation after interacting with the membrane oxygenator. To mitigate the risk of distal limb ischemia, an 8 Fr distal reperfusion cannula is inserted into the superficial femoral artery, creating an artificial bypass.

In a subset of patients with severely depressed contractility and/or concomitant mitral or aortic regurgitation, VA-ECMO may increase afterload and LV end diastolic pressure. In such circumstances, LV unloading with IABP or Impella is mandatory. A recent study showed that among adults receiving VA-ECMO, LV unloading was associated with lower in-hospital mortality despite increased complications including hemolysis and cannulation site bleeding. Compared to VAD, LV unloading with IABP was associated with similar mortality and lower complication rates [49].

Evidence for benefit of VA-ECMO in CS is based on several non-randomized studies reporting survival range from 33.8% to 66.7% in AMI and CS with the use of VA-ECMO ranged [50,51].

Recent evidence supports favorable outcomes with timely insertion of VA-ECMO. In a large Japanese registry, a shorter time interval between cardiac arrest and VA-ECMO insertion in out-of-hospital resuscitated patients due to cardiac causes was an independent predictor of improved neurologic outcomes at 30 days [52].

Early placement of VA-ECMO in CS may also be beneficial in terms of 30-day all-cause mortality according to a study including CS patients (56% had post-AMI CS). The Shock-to-ECMO time was defined as the time interval between the onset of refractory CS and the time when the ECMO’s centrifugal pump was turned on. A short Shock-to-ECMO time (< 0.9 h) was associated with a 47% lower risk of 30-day mortality when compared to longer Shock-to-ECMO time (>2.2 h) [53].

## 9. Combining MCS Devices in CS Management

Data from registries indicate rare use of more than one MCS in a single patient. Among 12,077 patients with CS complicating AMI, 86.5% of patients received one MCS (IBAP, Impella, TendemHeart, ECMO, or LVAD) while 13.5% of patients underwent dual device placement or received another MCS device. The most frequent combinations were IBAP plus Impella (2.3%) and IBAP plus ECMO (1.1%) [44]. In contrast, LV “venting” or “unloading” may be necessary in the setting of increased afterload induced by VA-ECMO attributable to retrograde aortic flow. LV unloading can be achieved by adding another MCS, such as IBAP or Impella, or by surgical maneuvers such as atrial septostomy. VA-ECMO with LV venting may be associated with lower mortality rate when compared to VA-ECMO alone. In a recently published multicenter cohort study comparing VA-ECMO alone to VA-ECMO with Impella, LV unloading with the combined approach was associated with a 21% lower 30-day mortality [54]. In a sub-analysis of matched cohorts evaluating early LV decompression with Impella within 2 h of VA-ECMO initiation vs. delayed Impella (>2 h after VA-ECMO), the expedited combined strategy was associated with a lower 30-day mortality when compared to VA-ECMO alone, whereas the delayed strategy was not. Thus, in patients on VA-ECMO, early LV unloading may have a survival benefit [54]. However, the populations in which LV “venting” offers a clear benefit remain largely uncharacterized. Finally, a higher rate of complications (bleeding, vascular complications) is seen with the combined approach [54].

Selected ongoing MCS studies in CS are summarized in Table 3.

## 10. The Use of MCS in the Setting of CS Due to Mechanical Complications of AMI

The incidence of mechanical complications and associated CS following AMI has declined in recent years, but mortality rates remain high, between 10 and > 50% [7]. In the SHOCK trial registry, acute severe mitral regurgitation secondary to papillary muscle rupture was the most common complication (6.9%), followed by ventricular septal rupture (3.4%) and free wall rupture (1.4%) [55]. Initial medical stabilization efforts, including vasoactive drugs and mechanical ventilation, are standard care but are rarely sufficient, and surgical or percutaneous repair are often perused [56]. However, emergency surgery in a decompensated CS patient is associated with dismal prognosis. Thus, despite the limited experience and evidence with short-term circulatory support in this setting, MCS bridge to surgery to achieve hemodynamics stabilization and end-organ perfusion is preferred prior to definitive repair to potentially improve outcomes [57].

## 11. Structural Heart Interventions for Emergent Treatment of Patients with CS

Percutaneous transcatheter heart interventions are a viable alternative to surgery in acutely ill patients at extremely high operative risk. Minimally invasive procedures, with or without MCS support, of both mitral and aortic valves have been described in the setting of CS in small case series and registries, with high procedural success rate and improved outcomes [58,59,60]. However, the populations in those specific studies are highly heterogeneous regarding CS etiology and include only small samples of post-AMI CS patients with secondary mitral regurgitation (MR). In a nationwide analysis evaluating the outcomes of mitral edge-to-edge repair with MitraClip, only 18% of the matched cohort had AMI, and only 15.8% underwent revascularization [60]. According to registries focusing on post-MI MR, without CS, early MR repair with transcatheter edge-to-edge repair (TEER) may be beneficial [61].

Furthermore, in a small cohort of post-AMI CS cases with acute MR, patients who underwent TEER had similar clinical outcomes compared to patients without CS, provided hemodynamic stabilization was first achieved before MR repair [62], suggesting potential benefit of MR reduction in this subpopulation. However, the overall data supporting a favorable impact of structural heart interventions on outcomes in the setting post-AMI CS is extremely limited, and no evidence-based recommendations are currently available.

In summary, valves intervention in CS patients is an emerging treatment that will probably become more common in the future in selected patients with valve disease and CS.

## 12. Conclusions

Temporary circulatory support devices are an emerging and rapidly evolving technologies developed as an adjunctive treatment of CS. However, the evidence for survival benefit following their use remains scarce. Multiple platforms are commercially available, each with its own pros and cons. Thus, a high level of expertise is essential to effectively address issues such as device selection, timing of treatment initiation, device troubleshooting, post-implantation care, and weaning. In addition, further prospective randomized data are urgently needed to formulate effective MCS management strategies in order to maximize their potential benefit.

## Figures and Tables

**Figure 1 jcm-11-05241-f001:**
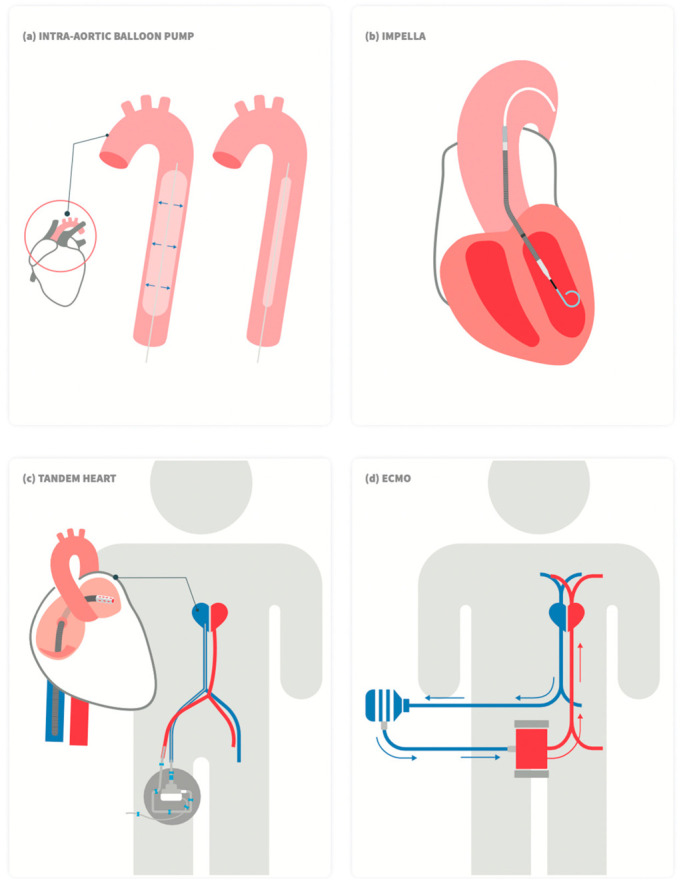
Schematic drawing of commercially available left ventricular percutaneous mechanical support devices. (**a**) Intra-aortic balloon pump, (**b**) Impella, (**c**)TandemHeart, and (**d**) Veno-arterial (VA) extracorporeal membrane oxygenation (ECMO).

**Table 1 jcm-11-05241-t001:** Shock stages: description and hemodynamics.

Stage	Description	Hemodynamics	Biochemical Markers
A At risk	No signs or symptoms of CS but at risk for CS development. May include patients with large acute myocardial infarction.	Normotensive (SBP ≥ 100 or normal for patient) If hemodynamics done: - Cardiac index ≥ 2.5 - CVP < 10 - PA sat ≥ 65%	Normal labs - Normal renal function - Normal lactic acid
B Beginning CS	A patient who has clinical evidence of relative hypotension or tachycardia without hypoperfusion.	SBP <90 **or** MAP <60 **or** >30 mmHg drop from baseline. - Pulse ≥ 100 - If hemodynamics done - Cardiac index ≥ 2.2 - PA sat ≥ 65%	- Normal lactate - Minimal renal function impairment - Elevated BNP
C Classic CS	A patient that manifests with hypoperfusion that requires intervention (inotrope, pressor, or mechanical support, including ECMO) beyond volume resuscitation to restore perfusion. These patients typically present with relative hypotension.	May include any of: SBP <90 **or** MAP <60 **or** >30 mmHg drop from baseline **and** drugs/device used to maintain BP above these targets Hemodynamics: - Cardiac index < 2.2 - PCWP >15 - RAP/PCWP ≥ 0.8 - PAPI < 1.85 - Cardiac power output ≤ 0.6	May include any of the following: - Lactate ≥2 - Creatinine doubling OR >50% drop in GFR - Increased LFTs - Elevated BNP
D Deteriorating	A patient that is similar to category C but is getting worse. They have failure to respond to initial interventions.	Any of Stage C **and**: Requiring multiple pressors OR addition of mechanical circulatory support devices to maintain perfusion	Any of Stage C and: Deteriorating
E Extrimis	A patient that is experiencing cardiac arrest with ongoing CPR and/or ECMO being supported by multiple interventions.	No SBP without resuscitation PEA or refractory VT/VF hypotension despite maximal support	“Trying to die” - CPR (A-modifier) - pH ≤7.2 - Lactate ≥5

**Table 2 jcm-11-05241-t002:** Characteristics of short-term mechanical circulatory support devices.

Device	IABP	Impella (2.5, CP, 5.0, 5.5, ECP)	TendemHeart LA-FA	Impella RP	TendemHeart RA-PA	VA-ECMO
Flow	-	2.5–5.5 L/min	Max 4 L/min	Max 4 L/min	Max 4 L/min	Max 7 L/min
Pump speed	-	Max 51,000 rpm	Max 7500 rpm	Max 33,000 rpm	Max 7500 rpm	Max 5000 rpm
Mechanism	Cardiac cycle timed balloon inflation-deflation	Axial flow continuous pump (LV to Ao)	Centrifugal flow continuous pump (LA to Ao)	Axial flow continuous pump (RA to PA)	Centrifugal flow continuous pump	Centrifugal flow continuous pump (RA to Ao)
Cannula size	7–8F arterial	9–14F arterial	12–19F arterial 21F venous	22F venous	29F venous	14–19F arterial 17–21F venous
Insertion	Femoral artery	Femoral artery	Femoral vein Femoral artery	Femoral vein	Internal jugular vein	Femoral vein Femoral artery
LV unloading	+	+ to +++	++	−	−	−
RV unloading	−	−	-	+	+	++
Cardiac power	−	↑↑	↑↑	-	-	↑↑
Afterload	↓	↓↓	↑	-	-	↑↑
Coronary perfusion	↑	↑	-	-	-	-
Complications	Cannula migration from LA to RA, tamponade, stroke, limb ischemia	Bleeding, hemolysis, vascular injuries, stroke, aortic valve injury	Dislodging of cannula, limb ischemia, femoral arteriovenous fistula, thromboembolism	Bleeding, hemolysis, vascular injuries, RV perforation, arrhythmia	Dislodging of cannula, vascular injury	Bleeding, thromboembolism, limb ischemia, renal failure, infections (including access site) lung edema, bleeding and hemoptysis
Contraindications	Severe aortic regurgitation, severe peripheral vascular disease precluding use	Severe peripheral vascular disease precluding use, LV thrombus, mechanical aortic valve, severe RV failure, aortic valve orifice area of 0.6 cm^2^ or less	Ventricular septal defect, significant aortic regurgitation	Inferior vena cava filter, severe tricuspis and/or pulmonic valve stenosis, mechanical right sided valves, thrombi in vena cava, right atrium	Ventricular septal defect	Expected lack of benefit (short life expectancy, terminal illness)

**Table 3 jcm-11-05241-t003:** On-going MCS trials in CS patients.

Study	Description	Sample Size	Primary Endpoint
Study on Early Intra-aortic Balloon Pump Placement in Acute Decompensated Heart Failure Complicated by Cardiogenic Shock (Altshock-2)	Patients will be randomized 1:1 to early IABP (within six hours of onset of cardiogenic shock) versus standard of care (vasoactive therapy).	200	60-day patients’ survival or successful bridge to heart replacement therapy.
Danish-German Cardiogenic Shock Trial (DanGer Shock)	Patients will be randomized to receive conventional circulatory support or support with the Impella CP device for a minimum of 48 h and inotropic support if needed.	360	All-cause mortality
Acute Impact of the Impella CP Assist Device in Pts. With Cardiogenic Shock on the Patients Hemodynamic (JenaMACS)	Assessment of the acute hemodynamic effects following implantation of the IMPELLA CP cardiac support device	20	Surrogate endpoint
Impella CP With VA ECMO for Cardiogenic Shock (REVERSE)	VA-ECMO with Impella CP (LV venting) versus VA-ECMO alone in cardiogenic shock.	96	Surrogate endpoint
Transient Circulatory Support in Cardiogenic Shock (ALLOASSIST)	Transient circulatory support (VA-ECMO, Impella) vs. standard therapy.	240	In-hospital mortality (from inclusion day to day 180)
Assessment of ECMO in Acute Myocardial Infarction Cardiogenic Shock (ANCHOR)	VA-ECMO via the femoral route, with IABP in the contralateral femoral artery versus ESC guidelines management. (i.e., no devices).	400	Treatment failure at 30 days: Death in the ECMO group and death OR rescue ECMO in the control group
Extracorporeal Life Support in Cardiogenic Shock (ECLS-SHOCK)	Extracorporeal life support and revascularization (PCI or CABG; ECLS insertion should be performed preferably before revascularization) versus revascularization alone.	420	30-day mortality
ExtraCorporeal Membrane Oxygenation in the Therapy of Cardiogenic Shock (ECMO-CS)	VA-ECMO versus control in cardiogenic shock complicating myocardial infarction.	120	Composite of death from any cause, resuscitated circulatory arrest, and implantation of another mechanical circulatory support device within 30 days
Testing the Value of Novel Strategy and Its Cost Efficacy in Order to Improve the Poor Outcomes in Cardiogenic Shock (EUROSHOCK)	Early intervention with ECMO therapy vs. standard treatment (no ECMO).	428	All-cause mortality
ECMOsorb Trial-Impact of a VA-ECMO in Combination With CytoSorb in Critically Ill Patients With Cardiogenic Shock (ECMOsorb)	VA-ECMO and CytoSorb (An extracorporeal cytokine hemoadsorption system is integrated in the VA-ECMO circuit) vs. VA-ECMO without CytoSorb.	54	Surrogate endpoint

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
