# Peer review of "Mechanical Circulatory Support Devices for the Treatment of Cardiogenic Shock Complicating Acute Myocardial Infarction—A Review"

_jcm, 2022, doi:10.3390/jcm11175241_

Round 1

Reviewer 1 Report

include a table on major trials on various types of MCS in cardiogenic shock complicating AMI.  

Author Response

Include a table on major trials on various types of MCS in cardiogenic shock complicating AMI.  

A table describing major trials on various types of MCS has been included in the original manuscript (please refer to Table 3 Page 21).

Reviewer 2 Report

I appreciate the editor for giving me the opportunity to comment the manuscript submitted to the Journal of Clinical Medicine by Bruoha et al. This report reviews the mechanical circulatory support for cardiogenic shock complicating acute myocardial infarction. The paper is well organized and comprehensively described. I consider the paper is informative for the readers of this journal. However, there are some points to be noted before this paper is published.

In “Pathophysiology, diagnosis, and prognosis” section, the authors described the definition of CS and mentioned the SCAI staging system. The definition is that of “general” CS and not related to MCS. Considering the purpose of this manuscript (i.e., to outline the current evidence of MCS utilization during CS in 44 the setting of AMI), it would be better to mention about more MCS related classification of heart failure, such as INTERMACS profile. The author should discuss more in detail about definitions or classifications of CS focusing on “when is MCS needed?”.    

The cause of CS in AMI patients is not always AMI itself but often the mechanical complications of MI, such as ventricular septal rupture, pupillary muscle rupture, and LV free wall rupture. MCS plays important roles to rescue these patients. This situation should also be discussed in this review. In this situation, MCS is often used as a “bridge to surgery”. I suggest the authors to refer to the recently published paper of Impella used as a bridge to surgery in this review (Interact Cardiovasc Thorac Surg. 2022 Jul 9;35(2):ivac088).

The authors seem to have rather negative impression about the combination of MCS devises in CS management. It seems that the authors underestimate the significance of increased afterlaoad of LV by ECMO. As a fact, the authors did not include “lung edema” in the complications of VA-ECMO (Table 1). Lung congestion caused by increased LV afterload by ECMO could even result in pulmonary bleeding. I suggest the authors to refer to the paper published in Circulation: Circulaiton. 2020; 142: 2095-2106.

In the final paragraph, the authors discussed about TAVR in CS. However, there is little, or no information about TAVR in AMI patients. This paragraph should be deleted.

Finally, the “take-home message” of this review article is unclear. I suggest the authors to create the conclusion paragraph at the end of this manuscript.

Author Response

In “Pathophysiology, diagnosis, and prognosis” section, the authors described the definition of CS and mentioned the SCAI staging system. The definition is that of “general” CS and not related to MCS. Considering the purpose of this manuscript (i.e., to outline the current evidence of MCS utilization during CS in 44 the setting of AMI), it would be better to mention about more MCS related classification of heart failure, such as INTERMACS profile. The author should discuss more in detail about definitions or classifications of CS focusing on “when is MCS needed?”.    

Thank you very much, we have expanded the section of CS classification and elaborated further the definitions regarding CS (please refer to page 4). For the sake of completion, we also added a new table regarding the SCAI classification. Including the INTERMACS profiles classification, as suggested by the referee, would render the review more comprehensive covering not only the role of MCS in the newly decompensated cases, but also in the chronic heart failure patients. A new paragraph regarding the INTERMACS was added (please refer to page 5). Finally, the timing of MCS initiation is one of the most important clinical questions in the management of CS patients and deserves a detailed discussion.  We added a section on the optimal timing for MCS initiation (please refer to page 7).  

The cause of CS in AMI patients is not always AMI itself but often the mechanical complications of MI, such as ventricular septal rupture, pupillary muscle rupture, and LV free wall rupture. MCS plays important roles to rescue these patients. This situation should also be discussed in this review. In this situation, MCS is often used as a “bridge to surgery”. I suggest the authors to refer to the recently published paper of Impella used as a bridge to surgery in this review (Interact Cardiovasc Thorac Surg. 2022 Jul 9;35(2):ivac088). 

Thank you for the comment, although uncommon, CS secondary to mechanical complications in the setting of AMI is an important entity and deserve a dedicated section. Consequently, a new paragraph on MCS in the setting of mechanical complication was added to the manuscript (page 14). We also agree that “bridge to surgery” in these patients deserve more detailed explanation (included in the same paragraph). The reference suggested by the referee was of major help.

The authors seem to have rather negative impression about the combination of MCS devises in CS management. It seems that the authors underestimate the significance of increased afterlaoad of LV by ECMO. As a fact, the authors did not include “lung edema” in the complications of VA-ECMO (Table 1). Lung congestion caused by increased LV afterload by ECMO could even result in pulmonary bleeding. I suggest the authors to refer to the paper published in Circulation: Circulaiton. 2020; 142: 2095-2106. 

Thank you for the comment, pulmonary edema (and potentially subsequent pulmonary bleeding and hemoptysis) is an important complication of VA-ECMO and is often an indication for LV “venting”. This information was added to the manuscript, including to table 2.

In the final paragraph, the authors discussed about TAVR in CS. However, there is little, or no information about TAVR in AMI patients. This paragraph should be deleted. 

Thank you for the comment, the paragraph was deleted.

Finally, the “take-home message” of this review article is unclear. I suggest the authors to create the conclusion paragraph at the end of this manuscript. 

Thank you for the comment, a conclusion paragraph was added to the manuscript.

Reviewer 3 Report

In this review, Bruoha et al describe Mechanical Circulatory Support Devices for the Treatment of  Cardiogenic shock Complicating Acute Myocardial Infarction.

Here my comments:

Review abstract and conclusion are needed

Many sentences, especially in the introduction, need references.

The authors treated just the CS problem after AMI, but this is not the only condition when MCS are needed (ex bridge to CABG or bridge to decision..), so these other conditions in a review need to be cited

A general conclusion about the use of MCS is needed

In a review, a reader expects to read a complete dissertation of the SC classification while only that of SCAI has been described, which is true and is the most used but not the only one.

Author Response

Review abstract and conclusion are needed

Thank you for the comment, an abstract and a conclusions sections were added to the manuscript.

Many sentences, especially in the introduction, need references.

Thank you for the comment, several references were added to the manuscript, especially in the introduction.

The authors treated just the CS problem after AMI, but this is not the only condition when MCS are needed (ex bridge to CABG or bridge to decision..), so these other conditions in a review need to be cited

Thank you for the comment, other uses of MCS are equally important and we expanded several sections to include the use of MCS in the chronic heart failure patients using the INTERMACS profiles (please refer to page 5), the use of MCS as a bridge to decision (please refer to page 7) and the use of MCS in the setting of mechanical complications of AMI (a new paragraph was added; please refer to page 14).

A general conclusion about the use of MCS is needed

Thank you for the comment, a conclusion paragraph was added to the manuscript (please refer to page 15).

In a review, a reader expects to read a complete dissertation of the SC classification while only that of SCAI has been described, which is true and is the most used but not the only one.

Thank you for the comment, the original paragraph regarding was somewhat missing. We completely revised the paragraph “Pathophysiology, diagnosis, and prognosis” adding more useful definition criteria used in CS and we also expanded the discussion regarding SCAI classification. Furthermore, a new table (table 1) was added to the manuscript to provide more clarity.

Round 2

Reviewer 2 Report

The manuscript is appropriately revised in most of the part.

However, I am afraid that the previous paper I recommended is not sited in the revised manuscript.

Reviewer 3 Report

The authors have made a great effort and have greatly improved the manuscript making it easy to read, didactic and of great interest